# Trend and predictive psychosocial factors of persistent depression/non-depression in Chinese adolescents: A three-year longitudinal study

**Ming Liu[1]⊛, Fei Xie[2]⊛, Zongpei Dai[1], Qin Dai👤[1]\*, Xin-li Chi[3,4]\***

**1** Department of Medical Psychology, Army Medical University, Chong Qing, China, **2** Department of Nursing School, Army Medical University, Chong Qing, China, **3** School of Psychology, Shenzhen University, Shenzhen, China, **4** The Shenzhen Humanities & Social Sciences Key Research Bases of the Center for Mental Health, Shenzhen University, Shenzhen, China

⊛ These authors contributed equally to this work.
\* daiqin101@hotmail.com (QD); xinlichi@126.com (XLC)

**Data Availability Statement:** All data generated or analyzed during this study are available from Supplementary materials of this paper.

## Abstract

### Background

Depressive symptoms are common in adolescents, while its prevalence trend in adolescents is seldom reported. In addition, few studies focus on adolescents without depressive symptoms. This study aimed to reveal the trend and predictors of persistent depression/non-depression among Chinese adolescents over three-year period.

### Methods

Three-wave data were collected from T1 to T3 in a sample from six junior high schools in Shenzhen, China. The 20-item of Center for Epidemiologic Studies Depression Scale was used, with scoring ≥16 as depression.

### Results

Of all 1301 adolescents, 33.4%, 15.5% and 9.0% reported depressive symptoms at baseline, T2 and T3 respectively, while 66.6%, 47.7% and 39.6% reported non-depressive symptoms at three time-points. At the individual level, baseline positive identity and general positive youth development significantly increased persistent non-depression at T2 and T3. At the family level, baseline family harmony and communication consistently prevented persistent depression at T2 and T3, while family communication consistently promoted non-depression. At the school level, baseline awareness of no expectation from teacher and poor interpersonal relationship consistently increased persistent depression and prevented non-depression.

### Conclusions

The incidence of depression in adolescents aged 11–15 is not high. The rate of persistent non-depression is approximately 30% higher than that of persistent depression, and both

**Funding:** This research was funded by the key project of natural science foundation of Chongqing (cstc2020jcyj-zdxmX0009, recipient: Qin Dai), Humanities and Social Sciences of Ministry of Education Planning Foundation (Grant 23YJA190002, recipient: Xinli Chi), and Guangdong Philosophy and Social Science Planning 2023 Co-construction Project (Grant GD23XXL17, recipient: Xinli Chi). The funders had a role in study design, data collection and decision to publish, while they had no role in data analysis and preparation of the manuscript.

**Competing interests:** The authors have declared that no competing interests exist.

have a downward trend over years. Positive youth development and healthy family function are protective factors of depression, while school maladjustment decreases persistent non-depression. During early adolescence, more attention should be paid to family function and school maladjustment to promote non-depression in adolescents from a new broader view.

## Introduction

According to the latest UNICEF report on August 1$^{st}$ 2019 [1], depression emerges as a leading cause of adolescent illness and disability. Negative long-term functional and psychiatric outcomes of adolescents are quite common, including impairment in school, interpersonal relationships, substance abuse and suicide due to adolescent depression [2, 3]. As a subclinical stage of major depressive disorder, depressive symptoms, usually seen in the depressive illness and listed in standard diagnostic schedules and rating scales [4], are more pervasive in adolescents [5].

However, the prevalence of depressive symptoms varies in different studies among Chinese adolescents. A large cross-sectional survey found that 23.9% of the 10657 children and adolescents aged from 7 to 17 reported depressive symptoms in the southwest of China [6]. Based on a biennial survey from 25 provinces of China, researchers investigated 3056 adolescents between 10 and 15 years old, calculating the prevalence rate of depressive symptoms was 16.7% [7]. Until recently, a meta-analysis has showed that the prevalence of depressive symptoms in Chinese children and adolescents was associated with the regions where the studies were conducted: 17.8% in eastern, 23.7% in central, 22.7% in western, and 14.5% in northeast pasts of China [4], but just lacking the prevalence data in the southern China. Furthermore, although these studies investigated in a large sample size, they were often cross-sectional, not beneficial to trace the changing prevalence rate over time. Besides, previous studies have paid too much attention to adolescents with depressive symptoms [4, 5], while adolescents without depressive symptoms (non-depression) are often ignored. Although experiencing same adversity, same people suffer from depression, while most do not suffer from depression but manifest certain resilience, which could be used to prevent and cure depression. Adolescents without depressive symptoms may possess different personal diathesis (such as resilience) [8] and demographic-social characteristics (e.g., higher socio-economical status) which protect them from depressed mood. In fact, investigating those without depressive symptoms can provide quite a different and unique perspective to prevent adolescent from depression, enhance resilience and ensure better mental health in them [9]. Therefore, longitudinal studies to explore the prevalence and rate change of adolescents with/without depressive symptoms over time are needed.

Apart from the prevalence of adolescent depression, researchers also have been exploring risk and protective factors of adolescent depressive symptoms. Based on the ecological system theory [10], personal and environmental factors (family and school, et al.) contributed to mental health status together. Based on recent reviews, researchers have highlighted the integral role of ecological factors to the American National Institute of Mental Health Research Domain Criteria framework in predicting onset and maintenance of internalizing problems in youth [11]. Furthermore, the risk factors and protective factors of young people relating to the mental health outcomes of both direct and indirect exposure to climate change were reviewed through the lens of ecological system theory [12]. Thus, for adolescents, specifically, it could be hypothesized that individual, family, and school factors may predict the onset of their depression together (Fig 1). Considering the studies on adolescents without depressive symptoms are

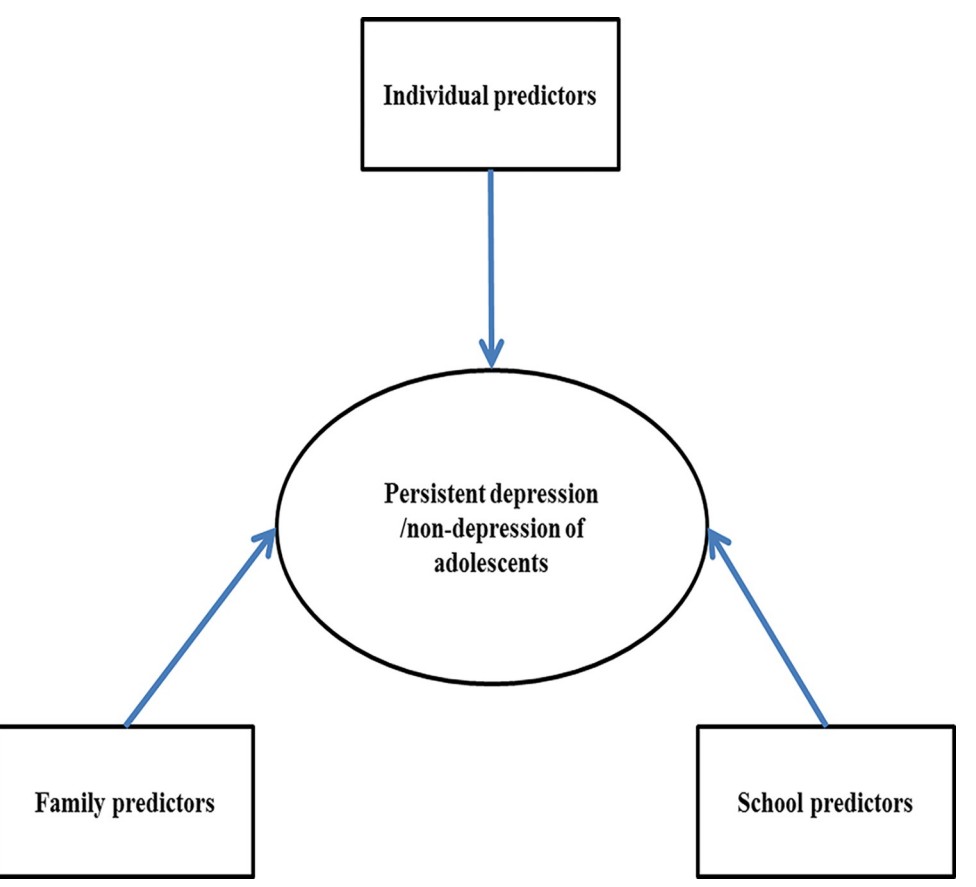

**Fig 1. Theoretical framework of this study based on ecological system theory.**

sparse, information about the factors contributing to non-depression in adolescents are even less, so we intended to explore psychosocial predictors (individual, family, and school factors) of persistent depression/non-depression in adolescents based on the theoretical framework of the ecological system theory [13], which is potentially important to figure out a way to better prevent adolescents from depression.

At the individual level, age and gender were correlated with adolescent depressive symptoms. It was found that the prevalence of adolescents with depressive symptoms above the cut-off (BDI-II > 13) was 11.9% at 13 years (girls: 17.1%; boys: 5.3%) and 10.8% at 17 years (girls: 14.7%; boys: 5.7%), which suggested that the prevalence of adolescents with BDI-II > 13 at both 13 and 17 years was higher in females compared with boy [13]. In China, one study showed that the prevalence of depression increased between 10–12 and 13–15 years old and then remained stable until 17 years old in the west China [6]. As to the gender difference, earlier studies revealed that girls were at risk for developing depressed disorder because they experienced more challenges in early adolescence than boys did [14, 15]. Besides the individual demographic factors described above, internal developmental assets and positive youth development (PYD) constructs of adolescents can help promote the healthy youth development and enhance their potential [16], suggesting that PYD may serve as a protective factor against adolescent depressive symptoms [17].

At the family level, parents being the main caretakers of adolescents may influence the attitudes, behaviors and socialization of adolescents [18]. Family relationship, family climate, parenting style and cohesion were significant factors correlating with adolescent depression [19–

21]. Additionally, adolescents from low socioeconomic families exhibit a higher rate of depression than those from mid-higher socioeconomic homes [22], suggesting that low family income may be a risk factor of adolescent depression. Besides, low parental education levels of parents also increase the risk of adolescent depression [23], and incomplete family structure had negative effects on the adolescent mental health [24]. For example, Icelandic adolescents from single-parent households reported more depressive symptoms than those who lived in intact families [25].

At the school level, school is a fundamental context for adolescents to achieve cognitive, affective, and social development [26]. When difficulties emerge for meeting the expectations of the school context, adolescents will confront school maladjustment, i.e., a set of behavioral, social, and emotional difficulties that prevent them from meeting expectations of their schools [27]. For example, a Chinese survey found that poor academic achievement and academic stress had negative effects on depressive symptoms among middle and high school students [28]. A recent study showed that teacher-student warmth was related to lower levels of depression [29].

Building upon the existing literatures about adolescent depression, although factors from individual, family, and school levels were indicated separately, however, most studies only focused on solely one domain of the ecological system, and most of them were cross-sectional design. Specifically, some variables under Chinese culture context (one-child policy, migrant population between provinces) [30] did not include in previous survey. Thus, their combined effect on longitudinal prediction of depression in Chinese adolescent remains unclear. Notably, predictors of adolescent non-depression from ecological system are unexplored.

Based on the ecological system theory, the present study firstly tended to describe the prevalence and occurrence trend of adolescents with and without depressive symptoms based on a three-year longitudinal follow-up investigation in Shenzhen, China. Secondly, to explore each set of psychosocial factors (individual, family, and school predictors) of persistent depression/non-depression symptoms among Chinese adolescents. Specifically we tested the following hypotheses: (1) Adolescence is a vulnerable period for depressive symptoms, and the prevalence of depressive symptoms increased during the transition from childhood into adolescence [31, 32]. We hypothesized that the occurrence rate of adolescents with depressive symptoms would be high (Hypothesis 1), which would show an increasing trend (Hypothesis 2), and adolescents without depressive symptoms would show a decreasing trend (Hypothesis 3) longitudinally. (2) Psychosocial factors from individual, family, and school levels would be associated with adolescent persistent depression/non-depressive symptoms. We hypothesized that the protective and risk factors predicting persistent depression symptoms (Hypothesis 4) and persistent non-depression symptoms (Hypothesis 5) could be identified in our study.

## Methods

### Participants

Our data were from a large-scale longitudinal study among Chinese junior secondary school students in Shenzhen city of south China. Six local public schools were randomly selected between October and November in 2016, all of which belonged to medium socio-economical status. 1,544 students in Grade 7th from these schools providing data for the first wave (T1). Between October and November in 2017 and 2018, the same students (then in Grades 8th and 9th, respectively) were invited to participate in the second (T2) and third wave (T3) of data collection, 1531 and 1480 students completed the questionnaire, respectively. Compared to the Grade 7th and 8th students, fewer students completed the survey in Grade 9th, partially due to the fact that some students had no household registration of Shenzhen and had to transfer back to schools at their hometowns to prepare for and attend the senior high school entrance

examination. Finally, 1301 participants aged 11–15 years-old were matched successfully with each unique school number, implying an acceptable attrition rate of 15.7%. Comparison between 243 dropouts and 1301 formal sample found no significant differences on demographic and baseline characteristics between them, which indicated that attrition did not bias the results (S1 Table). The mean age of the participants at wave 1 was 12.46 (SD = .63), and 666 of who were boys (51.19%).

## Measures

**Depression symptoms.**   The 20-item of Center for Epidemiologic Studies Depression Scale (CES-D) [33] was utilized to assess students' depression symptoms. Participants were required to assess how often they have experienced the listed symptoms of depression in the previous week using a four-point Likert scale (0 = none or almost none, 3 = often or almost often; α = .85, .88 and .87 at three waves). The sum of scores range from 0 to 60, higher score indicates a higher level of depression symptoms. Scoring less than 16 was classified as non-depression, scoring 16 or above was considered as depression in our study [34].

**Positive youth development (PYD).**   The 44-item of the Chinese Positive Youth Development Scale (CPYDS) was used to measure four PYD constructs [35], including cognitive-behavioral competence (nine items), positive identity (six items), prosocial attributes (six items), and general PYD qualities (23 items). The CPYDS was assessed by a six-point Likert scale (1 = strongly disagree, 6 = strongly agree; α = .95, .96 and .95 in three waves). The average total score of CPYDS was adopted as the indicator of positive qualities of adolescents mentioned above. Higher score represents a higher level of positive qualities.

**Family functioning.**   The 9-item of the Chinese Family Assessment Instrument (CFAI) was adopted to assess the general family function [36], especially in the family mutuality (3 items), family communication (3 items), and family conflicts (3 items). Each item was assessed by a 5-point Likert scale (1 = very similar, 5 = very dissimilar; α = .86, .89 and .85 at three waves). The mean scores of the CFAI were calculated, with higher average scores indicating healthier family functioning.

**School maladjustment.**   The 9-item of the Risky Behavior Questionnaire for Adolescent (RBQ-A) was utilized to measure the frequency with which the participant engaged in a range of maladaptive behaviors in school [37], including classroom problem behavior (2 items), academic fraud (3 items), awareness of no expectation from teachers (2 items), and having poor interpersonal relationship (2 items). Each item was assessed by a 5-point Likert scale (1 = never, 5 = always, α = .79, .80 and .78 at three waves). A higher total score represents a higher level of school maladjustment.

**Socio-demographic correlates.**   Participants were invited to report their age, gender (1 = male, 2 = female) and whether they were from a one-child family (1 = one child, 2 = non-one child), as well as whether a migrant student (1 = yes, 2 = no). They also reported their parents' education levels (1 = middle school or lower, 2 = high school or college, 3 = graduate, and 4 = above graduate) and whether they grew up in an intact family (1 = intactness, 2 = divorced parents, 3 = single parent family, 4 = others). Finally, they reported their per capita monthly family income (RMB) as indices (1 < 1,000, 2 = between 1,000 and 1,999, 3 = between 2,000 and 2,999, 4 = between 3,000 and 3,999, 5 = between 4,000 and 4,999, 6 = between 5,000 and 5,999, and 7 = ≥ 6,000).

## Procedure

The study and data collection received approval from the surveyed schools and the Human Research Ethics Committee of Shenzhen University. All methods were performed in

accordance with the relevant guidelines and regulations. All students were informed of the study purpose and their data would be analyzed and published with personal information kept in strict confidentiality. Before the data collection, the written informed consent of participants was obtained from both students and their parents. Three waves of questionnaire surveys (including the 20-item CES-D, CPYDS, CFAI, the 9-item subscale of RBQ-A, and socio-demographic information) were administered by two trained psychological graduate students in classroom settings with standardized instructions; one graduate student introduced the purpose of the study, and the other helped maintain class order. Students were required to sit separately to complete the above questionnaires, without talking or discussing with each other, which lasted about 20 minutes. Two trained instructors were present throughout the process to answer questions from participants. After survey, students were debriefed about the study and received incentives (e.g., notebooks, books, pens) from investigators.

## Statistical analyses

According to the three-wave CES-D scores in our study, three groups were identified [34]: ① Persistent depression symptom, in which adolescents were constantly classified as reporting symptoms (CES-D ≥16) from Wave 1 to 3; ②Persistent non-depression, in which adolescents report no depression symptoms (CES-D <16) continuously from T1 to T3; ③ Mixed depression symptom, in which adolescents were classified as reporting symptoms (CES-D ≥16) once (i.e., mixed depression across 12 months) or no more than twice (i.e., mixed depression across 24 months) from Wave 1 to 3.

The specific procedures of statistical analyses were as follows: firstly, to describe the trends of the above three groups among Chinese adolescents from T1 to T3, especially to examine the occurrence rate at three waves. Secondly, to describe and compare the psychosocial factors between three groups at T2 and T3 with $\chi^2$ test and one-way ANOVA. Thirdly, multilevel logistic regression analyses (level 1: individual; level 2: family; level 3: school) were conducted to examine the over-time predictive effects of baseline variables on depression scores of persistent depression/non-depression at T2 and T3, respectively. In which, potential confounding variable were controlled at level 1 (baseline age, gender, whether have siblings, migrant status) and level 2 (baseline father's and mother's educational level, family intactness or not, monthly household income). Specifically, being classified as persistence of depression/non-depression T2 and T3 were input as the dependent variable (i.e., 1 = met the criterion, 0 = did not meet the criterion), respectively. For independent variables, baseline age, gender, whether have siblings, migrant status, and PYD were put in the first block (level 1); baseline father's and mother's educational level, family intactness or not, monthly household income and family functioning, were input in the second block (level 2); the level of school maladjustment at baseline was entered in the third block (level 3); and level of T2 PYD (level 4), T2 family functioning (level 5), and T2 school maladjustment (level 6) were put in the fourth, fifth, and sixth blocks. Poisson distributions were normally approximated to calculate the 95% Confidence Intervals (CI). SPSS Version 24.0 was used for data analysis. Statistical significance was set at p < .05.

## Results

### Trend of the persistent depression/non-depression among Chinese adolescents (Figs 2 and 3)

Based on the scores of CES-D (*scoring less than 16 was classified as non-depression*) [34], three groups (peristent depression, mixed depression, and persistent non-depression) (Persistent

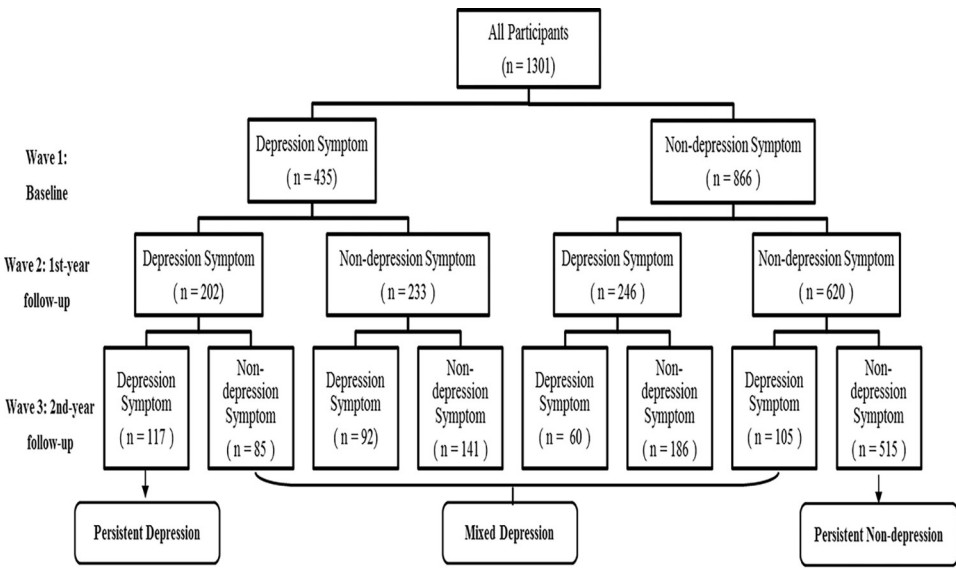

**Fig 2. Flowchart of participants at 3 waves of assessments.** Notes: Scoring less than 16 was classified as non-depression, while scoring 16 or above was considered as depression. Persistent Depression: Adolescents who reported depression symptoms (CES-D ≥16) at three assessments. Mixed depression: Adolescents who reported depression symptoms (CES-D ≥16) at one or two of the three assessments. Persistent Non-depression: Adolescents who did not report depression symptoms (CES-D <16) at three assessments.

depression symptom, in which adolescents were constantly classified as reporting symptoms (CES-D ≥16) from Wave 1 to 3. Persistent depression symptom, in which adolescents were constantly classified as reporting symptoms (CES-D ≥16) from Wave 1 to 3.) were identified. Figs 2 and 3 showed that 66.6% of 1301 participants did not report depression symptoms at baseline. The prevalence of persistent non-depression was 47.7% and 39.6% at T2 and T3 respectively, a downward trend of the persistent non-depression among Chinese adolescents year by year. A similar downward trend of the prevalence of persistent depression was also found in our study, the rates of which were 33.4%, 15.5% and 9.0% at baseline, T2 and T3 respectively.The results suggested that two-third adolescents exempted from depression symptoms cross-sectionally and three-fifth (60.4%) adolescents experienced a kind of depression longitudinally. Across three timepoints, the rate of persistent non-depression was approximately 30% higher than that of persistent depression, although both had a downward trend.

## Individual, Family and School Characteristics between three groups at 1st-year and 2nd-year follow-up (Table 1)

At 1st-year and 2nd-year follow-up, three groups significantly differed on total scores of CES-D, with highest scores in persistent depression group and lowest scores in persistent non-depression group. Persistent non-depression consistently reported less migrant students, higher education level of parents, higher scores of PYD, better family function, and less school maladjustment, while persistent depression group reported more sisters amd brothers. In addition, persistent non-depression reported more from intact family at 1st-year follow-up

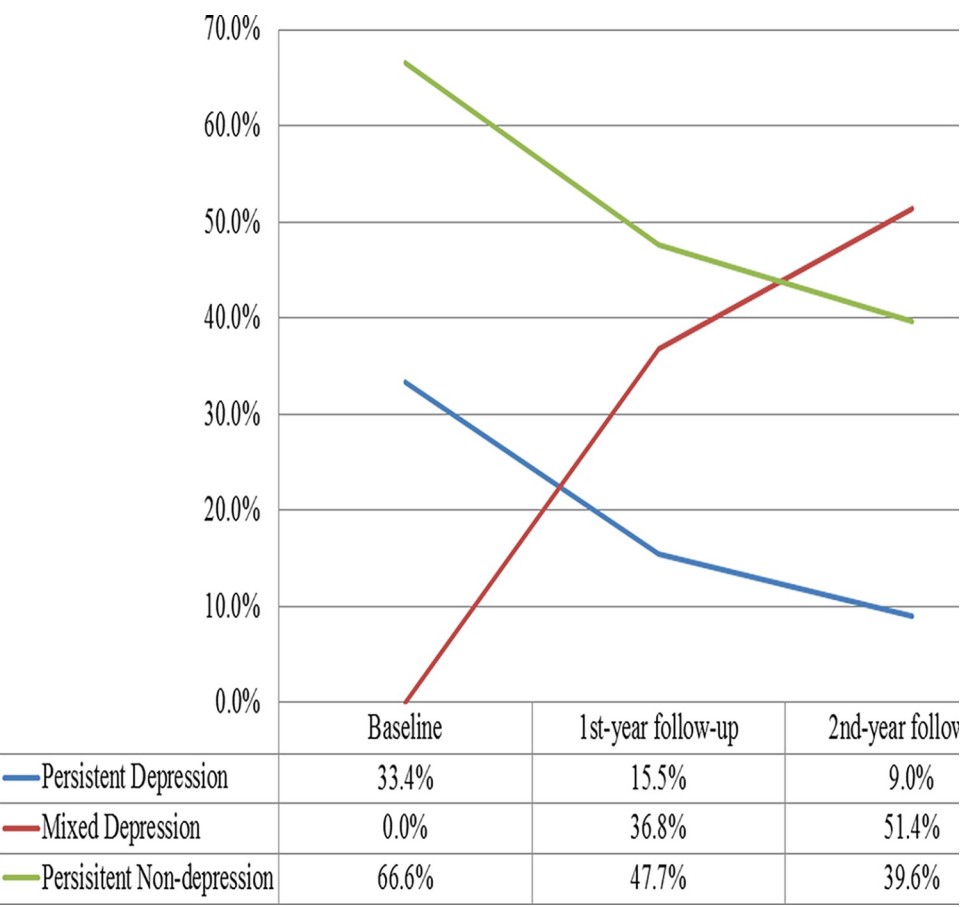

| | Baseline | 1st-year follow-up | 2nd-year follow |
|---|---|---|---|
| Persistent Depression | 33.4% | 15.5% | 9.0% |
| Mixed Depression | 0.0% | 36.8% | 51.4% |
| Persisitent Non-depression | 66.6% | 47.7% | 39.6% |

**Fig 3. Line chart of prevalence rates of three groups at 3 waves.**

and higher family income at 2nd-year follow-up. The results indicated a better personal, family and school condition in persistent non-depression group.

## Predictive factors of persistent depression among Chinese adolescents across one and two years (Tables 2 and 3)

Regarding predictors of persistent depression at one-year follow-up, Table 2 showed that above graduate education of father, and T1 family harmony and communication negatively predicted T2 adolescents' depression, while T1 awareness of no expectation from teacher and poor interpersonal relationshipin school positively predicted T2 adolescents' depression.

Regarding predictors of persistent depression at two-year follow-up, Table 3 showed that T1 personal positive identity and family harmony and communication negatively predicted T3 adolescents' depression, while T2 prosocial attributes and poor interpersonal relationship in school positively predicted T3 adolescents' depression.

The results indicated that family harmony and communication consistently protected adolescents from persistent depression, and poor interpersonal relationship in school consistently increased the occurrence of persistent depression in adolescent. In addition, personal positive identityand above graduate education of father protected adolescents from depression while prosocial attributes increased depressionoccasionally.

**Predictive factors of persistent non-depression among Chinese adolescents across one and two years (Tables 4 and 5).** Regarding predictors of persistent non-depression at one-

**Table 1. Individual, Family and School characteristics between *three groups* at 1st-year and *2nd*-year follow-up (N = 1301).**

| Predictors | | 1st-year follow-up | | $F/x^2$ | | 2nd-year follow-up | | $F/x^2$ |
|---|---|---|---|---|---|---|---|---|
| | Persistent depression | Mixed depression | Persistent non-depression | | Persistent depression | Mixed depression | Persistent non-depression | |
| | n/% (M/SD) n = 202 | n/% (M/SD) n = 479 | n/% (M/SD) n = 620 | | n/% (M/SD) n = 117 | n/% (M/SD) n = 669 | n/% (M/SD) n = 515 | |
| CES-D total scores | 25.28/7.43 | 16.42/9.17 | 7.95/4.16 | 534.85*** | 25.09/7.35 | 14.44/9.45 | 6.88/4.29 | 321.15*** |
| ***Individual factors*** | | | | | | | | |
| Age | 12.47/0.67 | 12.47/0.62 | 12.45/0.62 | 0.25 | 12.49/0.64 | 12.46/0.63 | 12.45/0.63 | 0.13 |
| Gender | | | | 2.74 | | | | 5.35 |
| Male | 92/46.5 | 248/52.1 | 326/53.2 | | 49/42.6 | 339/51.3 | 278/54.4 | |
| Female | 106/53.5 | 228/47.9 | 287/46.8 | | 66/57.4 | 322/48.7 | 233/45.6 | |
| Siblings | | | | 13.87** | | | | 8.93* |
| One Child | 54/26.7 | 194/40.5 | 251/40.7 | | 30/25.6 | 264/39.5 | 205/40.0 | |
| Non-one child | 148/73.3 | 285/59.5 | 366/59.3 | | 87/74.4 | 404/60.5 | 308/60.0 | |
| Migrant status | | | | 13.42** | | | | 7.11* |
| Migrant students | 50/25.0 | 96/20.1 | 89/14.4 | | 24/20.5 | 136/20.5 | 75/14.6 | |
| Local students | 150/75.0 | 381/79.9 | 529/85.6 | | 93/79.5 | 529/79.5 | 438/85.4 | |
| Positive Youth Development | 4.34/0.75 | 4.76/0.67 | 5.04/0.65 | 84.79*** | 4.39/0.74 | 4.98/0.79 | 5.32/0.63 | 87.59*** |
| ***Family factors*** | | | | | | | | |
| Family intactness | | | | 16.79* | | | | 7.86 |
| Intactness | 177/88.9 | 456/95.6 | 589/95.9 | | 104/90.4 | 628/94.6 | 490/95.9 | |
| Divorced parents | 11/5.5 | 12/2.5 | 13/2.1 | | 6/5.2 | 19/2.9 | 11/2.2 | |
| Single parent family | 8/4.0 | 6/1.3 | 10/1.6 | | 3/2.6 | 12/1.8 | 9/1.8 | |
| Others | 3/1.5 | 3/0.6 | 2/0.3 | | 2/1.7 | 5/0.8 | 1/0.2 | |
| Per capita monthly income in family (RMB) | | | | 19.10 | | | | 26.06* |
| <1,000 | 7/4.1 | 12/2.8 | 8/1.6 | | 5/5.2 | 16/2.7 | 6/1.4 | |
| 1,000–1,999 | 13/7.6 | 36/8.4 | 32/6.4 | | 11/11.3 | 44/7.5 | 26/6.3 | |
| 2,000–2,999 | 23/13.5 | 59/13.8 | 53/10.6 | | 16/16.5 | 77/13.1 | 42/10.1 | |
| 3,000–3,999 | 34/19.9 | 70/16.4 | 65/13.0 | | 19/19.6 | 95/16.2 | 55/13.3 | |
| 4,000–4,999 | 16/9.4 | 40/9.4 | 65/13.0 | | 4/4.1 | 66/11.2 | 51/12.3 | |
| 5,000–5,999 | 14/8.2 | 51/11.9 | 62/12.4 | | 6/6.2 | 74/12.6 | 47/11.3 | |
| ≥6,000 | 64/37.4 | 159/37.2 | 216/43.1 | | 36/37.1 | 215/36.6 | 188/45.3 | |
| Father's Education Level | | | | 20.70** | | | | 25.92*** |
| Middle school or lower | 80/44.2 | 164/36.0 | 160/28.3 | | 53/50.5 | 221/35.4 | 130/27.5 | |
| High school or college | 63/34.8 | 163/35.8 | 225/39.8 | | 32/30.5 | 234/37.5 | 185/39.2 | |
| Graduate | 29/16.0 | 80/17.6 | 117/20.7 | | 16/15.2 | 105/16.8 | 105/22.2 | |
| Above graduate | 9/5.0 | 48/10.5 | 63/11.2 | | 4/3.8 | 64/10.3 | 52/11.0 | |
| Mother's Education Level | | | | 17.47** | | | | 21.70** |
| Middle school or lower | 95/51.9 | 188/41.3 | 209/36.5 | | 63/60.0 | 256/40.7 | 173/36.3 | |
| High school or college | 55/30.1 | 166/36.0 | 206/33.2 | | 25/23.8 | 229/36.4 | 173/36.3 | |
| Graduate | 22/12.0 | 74/16.3 | 121/21.2 | | 13/12.4 | 105/16.7 | 99/20.8 | |
| Above graduate | 11/6.0 | 27/5.9 | 36/6.3 | | 4/3.8 | 39/6.2 | 31/6.5 | |
| Family Function | 3.61/0.84 | 3.97/0.78 | 4.23/0.72 | 54.18*** | 3.60/.76 | 4.07/.79 | 4.42/.63 | 73.31*** |
| ***School factor*** | | | | | | | | |
| School Maladjustment | 21.76/5.83 | 18.38/5.67 | 15.39/4.75 | 121.53*** | 22.04/6.39 | 16.44/5.88 | 14.36/4.62 | 96.43*** |

*p < 0.05

**p < 0.01

***p < 0.001.

**Table 2. Multilevel logistic regression modeling results for persistent depression across one year.**

| Predictors | Model 1 B | Model 1 OR and 95% C.I. for OR | Model 2 B | Model 2 OR and 95% C.I. for OR | Model 3 B | Model 3 OR and 95% C.I. for OR |
|---|---|---|---|---|---|---|
| *Individual factors* | | | | | | |
| Age | -0.11 | 0.90 (0.67, 1.21) | -0.07 | 0.93 (0.68, 1.27) | -0.15 | 0.86 (0.63, 1.19) |
| Gender | | | | | | |
| Male | | 1 | | 1 | | 1 |
| Female | 0.29 | 1.34 (0.92, 1.95) | 0.33 | 1.39 (0.93, 2.06) | 0.37 | 1.45 (0.96, 2.19) |
| Siblings | | | | | | |
| One Child | | 1 | | 1 | | 1 |
| Non-one child | 0.57 | 1.76 (1.16, 2.67)** | 0.48 | 1.62 (1.02,2.57)* | 0.43 | 1.54 (0.96, 2.47) |
| Migrant status | | | | | | |
| Migrant students | | 1 | | 1 | | 1 |
| Local students | -0.36 | 0.69 (0.45, 1.08) | -0.20 | 0.85 (0.59,1.21) | -0.26 | 0.77 (0.48, 1.25) |
| Positive Youth Development (PYD) | | | | | | |
| Cognitive Behavioral Competence | -0.13 | 0.88 (0.63, 1.24) | -0.16 | 0.85 (0.59, 1.21) | -0.14 | 0.87 (0.61, 1.24) |
| Prosocial Attributes | -0.12 | 0.88 (0.68, 1.15) | -0.17 | 0.84 (0.64, 1.11) | -0.10 | 0.90 (0.68, 1.19) |
| Positive Identity | -0.30 | 0.77 (0.57, 0.97)* | -0.25 | 0.78 (0.59, 1.03) | -0.25 | 0.78 (0.59,1.04) |
| General PYD Qualities | -0.39 | 0.67 (0.45, 1.01) | -0.06 | 0.94 (0.58, 1.51) | 0.07 | 1.07 (0.66, 1.73) |
| *Family factors* | | | | | | |
| Family intactness | | | | | | |
| Intactness | | | | 1 | | 1 |
| Divorced parents | | | 0.75 | 2.11 (0.81, 5.51) | 0.75 | 2.10 (0.80, 5.54) |
| Single parent family | | | 0.91 | 2.49 (0.72, 5.64) | 1.12 | 2.89 (0.89, 5.98) |
| Others | | | 0.73 | 2.15 (0.63, 5.19) | 2.25 | 2.51 (0.79, 5.70) |
| Per capita monthly income in family (RMB) | | | | | | |
| <1,000 | | | | 1 | | 1 |
| 1,000–1,999 | | | -0.17 | 0.84 (0.22,3.23) | -0.07 | 0.93(0.23, 3.74) |
| 2,000–2,999 | | | -0.01 | 0.98 (0.28, 3.46) | -0.03 | 0.97 (0.26, 3.58) |
| 3,000–3,999 | | | 0.45 | 1.57 (0.47, 5.26) | 0.55 | 1.74 (0.49, 4.12) |
| 4,000–4,999 | | | -0.04 | 0.96 (0.27, 3.45) | 0.08 | 1.08 (0.28, 4.01) |
| 5,000–5,999 | | | -0.24 | 0.79 (0.21, 2.95) | -0.11 | 0.89 (0.23, 3.54) |
| ≥6,000 | | | 0.32 | 1.37 (0.42, 4.48) | 0.34 | 1.40 (0.40, 4.84) |
| Father's Education Level | | | | | | |
| Middle school or lower | | | | 1 | | 1 |
| High school or college | | | -0.01 | 0.99 (0.61, 1.59) | -0.05 | 0.95 (0.58, 1.55) |
| Graduate | | | -0.03 | 0.97 (0.47, 2.00) | 0.03 | 1.03 (0.49, 2.14) |
| Above graduate | | | -1.44 | 0.24 (0.07, 0.80)* | -1.37 | 0.25 (0.07, 0.86)* |
| Mother's Education Level | | | | | | |
| Middle school or lower | | | | 1 | | 1 |
| High school or college | | | -0.30 | 0.74 (0.45, 1.21) | -0.29 | 0.75 (0.45, 1.24) |
| Graduate | | | 0.05 | 1.06 (0.49, 2.27) | -0.04 | 0.96 (0.43, 2.11) |
| Above graduate | | | 0.94 | 2.56 (0.83, 7.88) | 0.94 | 2.56 (0.81, 8.14) |
| Family Function | | | | | | |
| Family mutuality | | | 0.05 | 1.05 (0.79, 1.39) | 0.04 | 1.04 (0.78, 1.39) |
| Family harmony | | | -0.31 | 0.73 (0.57, 0.94)* | -0.27 | 0.77 (0.59, 0.99)* |
| Family communication | | | -0.36 | 0.69 (0.54, 0.89)** | -0.37 | 0.69 (0.54, 0.88)** |
| *School factor* | | | | | | |

(*Continued*)

**Table 2.** (Continued)

| Predictors | Model 1 | | Model 2 | | Model 3 | |
|---|---|---|---|---|---|---|
| | $B$ | OR and 95% C.I. for OR | $B$ | OR and 95% C.I. for OR | $B$ | OR and 95% C.I. for OR |
| School Maladjustment | | | | | | |
| Classroom problematic behaviour | | | | | -0.05 | 0.95 (0.81, 1.12) |
| Academic fraud | | | | | 0.09 | 1.10 (0.99, 1.22) |
| Awareness of no expectation from teacher | | | | | 0.11 | 1.11 (1.00, 1.23)* |
| Poor interpersonal relationship | | | | | 0.14 | 1.15 (1.01, 1.31)* |
| $\chi^2$ | 75.15*** | | 124.57*** | | 146.64*** | |
| -2 × log likelihood | 767.12 | | 717.69 | | 695.63 | |
| Cox & Snell $R^2$ | 0.07 | | 0.12 | | 0.14 | |
| Nagelkerke $R^2$ | 0.13 | | 0.21 | | 0.24 | |

*$p < 0.05$

**$p < 0.01$

***$p < 0.001$, PYD = Positive Youth Development.

year follow-up, Table 4 showed that T1awareness of no expectation from teacher and poor interpersonal relationshipin school negatively predicted T2 adolescents' non-depression, while local students, T1 personal positive identity and general positive youthdevelopment qualities, good family communication positively predicted T2 adolescents' non-depression.

Regarding predictors of persistent non-depression at two-year follow-up, Table 5 showed that T1 and T2 awareness of no expectation from teacher and poor interpersonal relationshipin school, and T2 cognitive behavioral competencenegatively predicted T3 adolescents' non-depression, while local students, T1 and T2 personal general positive youth development qualities, and T1 family communication, and T2 personal positive identity positively predicted T3 adolescents' non-depression.

The results indicated that awareness of no expectation from teacher and poor interpersonal relationship in school consistently prevented adolescents from persistent non-depression, while local students, personal positive identity and general positive youth development qualities, and family communication promoted persistent non-depression in adolescents. In addition, personal cognitive behaviral competencedecreased non-depression of adolescents incidentally.

## Discussions

In the current study, one-third (33.4%) adolescents reported depressive symptoms (CES-D scores ≥16) at baseline. This rate is relatively lower compared with adolescents from Hubei province of China, which found that 35.11% of urban students aged 11–15 years-old displayed depressive symptoms (CES-D scores ≥20) [38]. In addition, this rate is also lower compared with migrant employment (45%, CES-D scores ≥16) in Japan [39]. Comparatively, this rate is higher than college students (18.2% in men and 27.5% in women (CES-D scores ≥16)) [40]. This finding suggests that the rate of cross-sectional depression in adolescents aged 11–15 years-old is not high, which is different from our hypothesis (Hypothesis 1). The reason may be that less migrant students, higher education level of parents, higher scores of PYD, better family function, and less school maladjustment in the current sample protected them from depression. Further analysis showed that three-fifth (60.4%) adolescents experienced a kind of depression (persistent depression or mixed depression) longitudinally. In other words, more

**Table 3. Multilevel logistic regression modeling results for persistent depression across two years.**

| Predictors | Model 1 B | Model 1 OR and 95% C.I. for OR | Model 2 B | Model 2 OR and 95% C.I. for OR | Model 3 B | Model 3 OR and 95% C.I. for OR | Model 4 B | Model 4 OR and 95% C.I. for OR | Model 5 B | Model 5 OR and 95% C.I. for OR | Model 6 B | Model 6 OR and 95% C.I. for OR |
|---|---|---|---|---|---|---|---|---|---|---|---|---|
| ***T1*** | | | | | | | | | | | | |
| ***Individual factors*** | | | | | | | | | | | | |
| Age | -0.07 | 0.93 (0.64,1.35) | -0.03 | 0.97 (0.65,1.44) | -0.13 | 0.88 (0.58,1.32) | -0.18 | 0.84 (0.55,1.27) | -0.16 | 0.85 (0.56,1.31) | -0.22 | 0.80 (0.52,1.25) |
| Gender | | | | | | | | | | | | |
| Male | | 1 | | 1 | | 1 | | 1 | | 1 | | 1 |
| Female | 0.33 | 1.38 (0.86,2.23) | 0.44 | 1.56 (0.94,2.59) | 0.49 | 1.64 (0.97,2.78) | 0.44 | 1.56 (0.90,2.70) | 0.43 | 1.54 (0.89,2.69) | 0.56 | 1.74 (0.97,3.15) |
| Siblings | | | | | | | | | | | | |
| One child | | 1 | | 1 | | 1 | | 1 | | 1 | | 1 |
| Non-one child | 0.60 | 1.82 (1.06,3.12) * | 0.27 | 1.31 (0.72,2.39) | 0.22 | 1.25 (0.68,2.31) | 0.18 | 1.19 (0.63,2.26) | 0.19 | 1.21 (0.64,2.29) | 0.07 | 1.07 (0.54,2.12) |
| Migrant status | | | | | | | | | | | | |
| Migrant students | | 1 | | 1 | | 1 | | 1 | | 1 | | 1 |
| Local students | 0.02 | 1.02 (0.56,1.83) | 0.28 | 1.33 (0.70,2.51) | 0.23 | 1.26 (0.66,2.42) | 0.08 | 1.08 (0.56,2.10) | 0.07 | 1.07 (0.55,2.08) | -0.01 | 0.99 (0.49,1.99) |
| Positive Youth Development (PYD) | | | | | | | | | | | | |
| Cognitive Behavioral Competence | 0.17 | 1.19 (0.78,1.81) | 0.10 | 1.10 (0.70,1.73) | 0.16 | 1.17 (0.75,1.83) | 0.19 | 1.21 (0.75,1.95) | 0.18 | 1.20 (0.75,1.93) | 0.23 | 1.26 (0.76,2.10) |
| Prosocial Attributes | -0.20 | 0.82 (0.59,1.13) | -0.27 | 0.77 (0.54,1.08) | -0.19 | 0.82 (0.58,1.17) | -0.12 | 0.89 (0.61,1.29) | -0.11 | 0.89 (0.61,1.30) | -0.15 | 0.87 (0.59,1.28) |
| Positive Identity | -0.51 | 0.60 (0.43,0.83) ** | -0.49 | 0.61 (0.43,0.87) ** | -0.48 | 0.62 (0.44,0.88) ** | -0.49 | 0.61 (0.43,0.88) ** | -0.50 | 0.61 (0.42,0.87) ** | -0.63 | 0.54 (0.36,0.79) ** |
| General PYD Qualities | -0.38 | 0.69 (0.42,1.13) | 0.11 | 1.12 (0.61,2.04) | 0.25 | 1.29 (0.69,2.39) | 0.39 | 1.47 (0.76,2.85) | 0.39 | 1.47 (0.76,2.86) | 0.40 | 1.49 (0.74,3.00) |
| ***Family factors*** | | | | | | | | | | | | |
| Family intactness | | | | | | | | | | | | |
| Intactness | | | | 1 | | 1 | | 1 | | 1 | | 1 |
| Divorced parents | | | 0.33 | 1.39 (0.40,4.78) | 0.36 | 1.44 (0.41,5.02) | 0.21 | 1.24 (0.35,4.41) | 0.12 | 1.13 (0.31,4.08) | -0.02 | 0.98 (0.25,3.90) |
| Single parent family | | | 0.61 | 1.85 (0.39,8.66) | 0.94 | 2.56 (0.56,11.63) | 0.60 | 1.83 (0.37,9.06) | 0.61 | 1.84 (0.37,9.16) | 0.85 | 2.34 (0.48,11.43) |
| Others | | | 0.85 | 2.34 (0.19,28.26) | 1.70 | 5.46 (0.45,66.15) | 1.80 | 6.06 (0.42,88.37) | 1.78 | 5.91 (0.41,86.06) | 1.20 | 3.33 (0.18,60.74) |
| Per capita monthly income in family (RMB) | | | | | | | | | | | | |
| <1,000 | | | | 1 | | 1 | | 1 | | 1 | | 1 |
| 1,000–1,999 | | | -0.38 | 0.68 (0.16,3.00) | -0.23 | 0.79 (0.17,3.74) | -0.25 | 0.78 (0.17,3.62) | -0.31 | 0.74 (0.16,3.46) | -0.44 | 0.64 (0.14,3.07) |

(*Continued*)

**Table 3.** (Continued)

| Predictors | Model 1 B | Model 1 OR and 95% C.I. for OR | Model 2 B | Model 2 OR and 95% C.I. for OR | Model 3 B | Model 3 OR and 95% C.I. for OR | Model 4 B | Model 4 OR and 95% C.I. for OR | Model 5 B | Model 5 OR and 95% C.I. for OR | Model 6 B | Model 6 OR and 95% C.I. for OR |
|---|---|---|---|---|---|---|---|---|---|---|---|---|
| 3,000–3,999 | | | -0.06 | 0.94 (0.25,3.57) | 0.10 | 1.11 (0.27,4.57) | -0.11 | 0.90 (0.22,3.69) | -0.12 | 0.88 (0.21,3.67) | -0.46 | 0.63 (0.15,2.68) |
| 4,000–4,999 | | | -1.21 | 0.30 (0.06,1.48) | -1.08 | 0.34 (0.06,1.83) | -1.23 | 0.29 (0.05,1.59) | -1.28 | 0.28 (0.05,1.52) | -1.52 | 0.22 (0.04,1.31) |
| 5,000–5,999 | | | -0.79 | 0.46 (0.10,2.08) | -0.58 | 0.56 (0.11,2.75) | -0.69 | 0.50 (0.10,2.45) | -0.76 | 0.47 (0.09,2.33) | -1.18 | 0.31 (0.06,1.62) |
| ≥6,000 | | | -0.13 | 0.88 (0.24,3.21) | -0.08 | 0.93 (0.23,3.71) | -0.16 | 0.85 (0.22,3.32) | -0.20 | 0.82 (0.21,3.23) | -0.43 | 0.65 (0.16,2.64) |
| Father's Education Level | | | | | | | | | | | | |
| Middle school or lower | | | | 1 | | 1 | | 1 | | 1 | | 1 |
| High school or college | | | -0.18 | 0.84 (0.46,1.54) | -0.21 | 0.81 (0.44,1.50) | -0.14 | 0.87 (0.47,1.63) | -0.14 | 0.87 (0.46,1.63) | -0.18 | 0.83 (0.43,1.61) |
| Graduate | | | 0.04 | 1.04 (0.42,2.57) | 0.17 | 1.18 (0.47,2.94) | 0.16 | 1.18 (0.46,3.03) | 0.20 | 1.22 (0.47,3.17) | -0.09 | 1.09 (0.41,2.92) |
| Above graduate | | | -1.69 | 0.18 (0.03,1.13) | -1.54 | 0.22 (0.04,1.33) | -1.41 | 0.24 (0.04,1.54) | -1.40 | 0.25 (0.04,1.60) | -1.54 | 0.21 (0.03,1.53) |
| Mother's Education Level | | | | | | | | | | | | |
| Middle school or lower | | | | 1 | | 1 | | 1 | | 1 | | 1 |
| High school or college | | | -0.61 | 0.54 (0.28,1.04) | -0.64 | 0.53 (0.27,1.03) | -0.65 | 0.52 (0.27,1.03) | -0.64 | 0.53 (0.27,1.05) | -0.62 | 0.54 (0.26,1.11) |
| Above graduate | | | -0.08 | 0.93 (0.19,4.60) | -0.20 | 0.82 (0.16,4.18) | -0.33 | 0.72 (0.14,3.67) | -0.33 | 0.72 (0.14,3.70) | -0.68 | 0.51 (0.08,3.12) |
| Family Function Baseline | | | | | | | | | | | | |
| Family mutuality | | | -0.05 | 0.96 (0.68,1.34) | -0.08 | 0.93 (0.65,1.31) | -0.04 | 0.96 (0.67,1.38) | -0.04 | 0.96 (0.67,1.38) | -0.02 | 0.98 (0.66,1.44) |
| Family harmony | | | -0.45 | 0.64 (0.48,0.87)** | -0.37 | 0.69 (0.50,0.95)* | -0.44 | 0.64 (0.46,0.90)** | -0.42 | 0.66 (0.47,0.92)* | -0.47 | 0.63 (0.44,0.90)* |
| Family communication | | | -0.36 | 0.70 (0.51,0.95)* | -0.38 | 0.69 (0.50,0.94)* | -0.43 | 0.65 (0.46,0.91)* | -0.41 | 0.66 (0.47,0.93)* | -0.48 | 0.62 (0.43,0.89)* |
| *School factors* | | | | | | | | | | | | |
| School Maladjustment | | | | | | | | | | | | |
| Classroom problem behaviour | | | | | 0.01 | 1.01 (0.82,1.24) | 0.02 | 1.02 (0.83,1.27) | 0.02 | 1.02 (0.83,1.27) | -0.10 | 0.91 (0.73,1.14) |
| Academic fraud | | | | | 0.09 | 1.10 (0.97,1.25) | 0.09 | 1.10 (0.96,1.25) | 0.10 | 1.11 (0.97,1.26) | 0.13 | 1.14 (0.99,1.32) |
| Awareness of no expectation from teacher | | | | | 0.10 | 1.11 (0.98,1.26) | 0.10 | 1.11 (0.97,1.27) | 0.10 | 1.10 (0.97,1.26) | 0.12 | 1.12 (0.97,1.30) |
| Poor interpersonal relationship | | | | | 0.17 | 1.18 (1.01,1.39)* | 0.20 | 1.22 (1.03,1.44)* | 0.19 | 1.21 (1.03,1.43)* | 0.16 | 1.17 (0.98,1.40) |
| **T2** | | | | | | | | | | | | |

(*Continued*)

**Table 3.** (Continued)

| Predictors | Model 1 | | Model 2 | | Model 3 | | Model 4 | | Model 5 | | Model 6 | |
|---|---|---|---|---|---|---|---|---|---|---|---|---|
| | *B* | OR and 95% C.I. for OR | *B* | OR and 95% C.I. for OR | *B* | OR and 95% C.I. for OR | *B* | OR and 95% C.I. for OR | *B* | OR and 95% C.I. for OR | *B* | OR and 95% C.I. for OR |
| *Individual factors* | | | | | | | | | | | | |
| Positive Youth Development (PYD) | | | | | | | | | | | | |
| Cognitive Behavioral Competence | | | | | | | -0.00 | 0.99 (0.61,1.64) | 0.02 | 1.02 (0.62,1.67) | 0.02 | 1.02 (0.62,1.70) |
| Prosocial Attributes | | | | | | | 0.45 | 1.56 (0.99,2.46) | 0.46 | 1.58 (1.00,2.50) | 0.58 | 1.78 (1.09,2.88)* |
| Positive Identity | | | | | | | -0.29 | 0.75 (0.50,1.11) | -0.29 | 0.75 (0.50,1.12) | -0.32 | 0.73 (0.48,1.09) |
| General PYD Qualities | | | | | | | -0.81 | 0.45 (0.25,0.78)** | -0.75 | 0.47 (0.26,0.87)* | -0.56 | 0.57 (0.31,1.06) |
| *Family factors* | | | | | | | | | | | | |
| Family Function | | | | | | | | | | | | |
| Family mutuality | | | | | | | | | 0.02 | 1.02 (0.69,1.49) | -0.04 | 0.97 (0.65,1.44) |
| Family harmony | | | | | | | | | -0.15 | 0.86 (0.62,1.20) | -0.16 | 0.85 (0.60,1.21) |
| Family communication | | | | | | | | | -0.07 | 0.93 (0.66,1.32) | -0.02 | 0.98 (0.68,1.43) |
| *School factors* | | | | | | | | | | | | |
| School Maladjustment | | | | | | | | | | | | |
| Classroom problematic behaviour | | | | | | | | | | | 0.06 | 1.06 (0.86,1.32) |
| Academic fraud | | | | | | | | | | | -0.02 | 0.98 (0.86,1.11) |
| Awareness of no expectation from teacher | | | | | | | | | | | 0.08 | 1.09 (0.95,1.25) |
| Poor interpersonal relationship | | | | | | | | | | | 0.36 | 1.43 (1.21,1.68)*** |
| $\chi^2$ | 56.23** | | 106.01*** | | 125.13*** | | 151.52*** | | 152.91*** | | 190.80*** | |
| -2 × log likelihood | 527.52 | | 477.73 | | 458.62 | | 432.23 | | 430.84 | | 392.95 | |
| Cox & Snell $R^2$ | 0.05 | | 0.10 | | 0.12 | | 0.14 | | 0.14 | | 0.17 | |
| Nagelkerke $R^2$ | 0.12 | | 0.23 | | 0.27 | | 0.32 | | 0.32 | | 0.39 | |

*$p < 0.05$

**$p < 0.01$

***$p < 0.001$.

**Table 4. Multilevel logistic regression modeling results for persistent non-depression across one year.**

| Predictors | Model 1 | | Model 2 | | Model 3 | |
|---|---|---|---|---|---|---|
| | *B* | OR and 95% C.I. for OR | *B* | OR and 95% C.I. for OR | *B* | OR and 95% C.I. for OR |
| *Individual factors* | | | | | | |
| Age | 0.04 | 1.04 (0.84, 1.29) | 0.05 | 1.05 (0.84, 1.31) | 0.09 | 1.10 (0.87, 1.38) |
| Gender | | | | | | |
| Male | | 1 | | 1 | | 1 |
| Female | -0.13 | 0.88 (0.67, 1.15) | -0.13 | 0.88 (0.67, 1.16) | -0.12 | 0.89 (0.67, 1.18) |
| Siblings | | | | | | |
| One Child | | 1 | | 1 | | 1 |
| Non-one child | -0.02 | 0.98 (0.74, 1.29) | 0.10 | 1.10 (0.81, 1.50) | 0.17 | 1.18 (0.87, 1.62) |
| Migrant status | | | | | | |
| Migrant students | | 1 | | 1 | | 1 |
| Local students | 0.46 | 1.58 (1.13, 2.23)** | 0.35 | 1.42 (0.10, 2.02) | 0.45 | 1.56 (1.09, 2.24)* |
| Positive Youth Development (PYD) | | | | | | |
| Cognitive Behavioral Competence | -0.17 | 0.85 (0.64, 1.12) | -0.15 | 0.86 (0.64, 1.15) | -0.18 | 0.84 (0.62, 1.13) |
| Prosocial Attributes | 0.06 | 1.06 (0.86, 1.31) | 0.08 | 1.09 (0.87, 1.35) | 0.01 | 0.99 (0.79, 1.24) |
| Positive Identity | 0.31 | 1.36 (1.08, 1.71)** | 0.29 | 1.34 (1.06, 1.69)* | 0.24 | 1.27 (0.99,1.61)* |
| General PYD Qualities | 0.74 | 2.10 (1.48,2.99)*** | 0.48 | 1.62 (1.11, 2.34)* | 0.33 | 1.39 (0.95,2.03)* |
| *Family factors* | | | | | | |
| Family intactness | | | | | | |
| Intactness | | | | 1 | | 1 |
| Divorced parents | | | -0.32 | 0.72 (0.29, 1.83) | -0.22 | 0.80 (0.32, 2.03) |
| Single parent family | | | -0.17 | 0.85 (0.28, 2.52) | -0.26 | 0.77 (0.25, 2.37) |
| Others | | | -0.42 | 0.66 (0.06, 6.87) | -1.01 | 0.37 (0.04, 3.66) |
| Per capita monthly income in family (RMB) | | | | | | |
| <1,000 | | | | 1 | | 1 |
| 1,000–1,999 | | | 0.36 | 1.43 (0.49, 4.22) | 0.45 | 1.58 (0.52, 4.78) |
| 2,000–2,999 | | | 0.33 | 1.39 (0.49, 3.89) | 0.49 | 1.63 (0.57, 4.71) |
| 3,000–3,999 | | | 0.20 | 1.22 (0.44, 3.37) | 0.30 | 1.36 (0.48, 3.86) |
| 4,000–4,999 | | | 0.72 | 2.06 (0.73, 5.80) | 0.84 | 2.31 (0.79, 6.69) |
| 5,000–5,999 | | | 0.55 | 1.74 (0.61, 4.93) | 0.68 | 1.97 (0.67, 5.72) |
| ≥6,000 | | | 0.47 | 1.59 (0.59, 4.28) | 0.64 | 1.90 (0.69, 5.25) |
| Father's Education Level | | | | | | |
| Middle school or lower | | | | 1 | | 1 |
| High school or college | | | 0.15 | 1.16 (0.82, 1.65) | 0.24 | 1.27 (0.88, 1.83) |
| Graduate | | | 0.19 | 1.21 (0.72, 2.03) | 0.21 | 1.23 (0.72, 2.10) |
| Above graduate | | | 0.23 | 1.25 (0.64, 2.48) | 0.28 | 1.32 (0.65, 2.67) |
| Mother's Education Level | | | | | | |
| Middle school or lower | | | | 1 | | 1 |
| High school or college | | | 0.06 | 1.07 (0.75, 1.52) | 0.03 | 1.03 (071, 1.48) |
| Graduate | | | 0.10 | 1.11 (0.65, 1.88) | 0.13 | 1.13 (0.66, 1.96) |
| Above graduate | | | -0.22 | 0.80 (0.37, 1.74) | -0.31 | 0.73 (0.33, 1.64) |
| Family Function | | | | | | |
| Family mutuality | | | -0.16 | 0.85 (0.68, 1.08) | -0.15 | 0.86 (0.68, 1.09) |
| Family harmony | | | 0.14 | 1.15 (0.94, 1.40) | 0.07 | 1.07 (0.87, 1.31) |
| Family communication | | | 0.36 | 1.44 (1.19, 1.73)*** | 0.36 | 1.43 (1.18, 1.73)*** |
| *School factor* | | | | | | |

*(Continued)*

**Table 4.** (Continued)

| Predictors | Model 1 | | Model 2 | | Model 3 | |
|---|---|---|---|---|---|---|
| | $B$ | OR and 95% C.I. for OR | $B$ | OR and 95% C.I. for OR | $B$ | OR and 95% C.I. for OR |
| School Maladjustment | | | | | | |
| Classroom problem behaviour | | | | | -0.03 | 0.97 (0.86, 1.09) |
| Academic fraud | | | | | -0.06 | 0.95 (0.87, 1.03) |
| Awareness of no expectation from teacher | | | | | -0.13 | 0.88 (0.81, 0.95)** |
| Poor interpersonal relationship | | | | | -0.17 | 0.84 (0.76, 0.94)** |
| $\chi^2$ | 101.82*** | | 132.43*** | | 180.25*** | |
| -2 × log likelihood | 1293.80 | | 1263.18 | | 1215.36 | |
| Cox & Snell R$^2$ | 0.10 | | 0.12 | | 0.16 | |
| Nagelkerke R$^2$ | 0.13 | | 0.16 | | 0.22 | |

*p < 0.05

**p < 0.01

***p < 0.001.

adolescents may experience depressive symptoms across time during junior secondary school study period. The reason may be that Chinese early adolescents in secondary schools, who are facing increasing academic challenges as the grade rises, along with high academic expectations from their parents. When they cannot obtain the expected academic performance, plus the lack of ability to reduce pressure effectively, they are more prone to depression [41–43]. Moreover, all six schools belonged to medium socio-economical status. However, Table 1 suggested that students came from families with different socio-economical statuses, with approximate 40% per capita monthly income in family (RMB) ≥6,000 and approximate 10% < 2000, which could represent students from low, medium, and high socio-economical statuses of families. Future studies are encouraged to include school of various socio-economical statuses to guarantee broader generalization of the results. Notably, seasonal affective disorder is a mood disorder that is characterized by depressive symptoms that occur at a specific time of year (typically fall or winter) with full remission at other times of year (typically spring or summer) [44]. The investigation time-points of this study were October to November during three successive years, which may be potentially influenced by the season change. Future study may choose other survey season to further verify the results.

Notably, two-third (66.6%) adolescents exempted from depression at baseline. Thus, it is unnecessary to be panic since that majority of adolescents are relatively healthy cross-sectional. In addition, the rate of both persistent depression and persistent non-depression decreased along with the time, which is corresponding to increased trend of mixed depression. Because that along with the time, some of adolescents may experience a certain level of depression, while others may remit from depressive symptoms. This is consistent with Hypothesis 3 (non-depression would show a decreasing trend) but different with Hypothesis 2 (depression would show an increasing trend). However, the rate of persistent non-depression was approximately 30% higher than that of persistent depression across three years. This novel finding is quite encouraging, indicating that more adolescents remain persistent non-depression (V.S. persistent depression) longitudinally, i.e., more youth exempt from depression compared to those suffer from depression symptoms. Together, our findings suggest that the rate of depression in adolescents aged 11–15 years-old is not high, and more adolescents keep persistent non-depression in general, which enhances Hypothesis 1 in a sense. This is a novel report referring

**Table 5. Multilevel logistic regression modeling results for persistent non-depression across two years.**

| Predictors | B | Model 1<br>OR and 95%<br>C.I. for OR | B | Model 2<br>OR and 95%<br>C.I. for OR | B | Model 3<br>OR and 95%<br>C.I. for OR | B | Model 4<br>OR and 95%<br>C.I. for OR | B | Model 5<br>OR and 95%<br>C.I. for OR | B | Model 6<br>OR and 95%<br>C.I. for OR |
|---|---|---|---|---|---|---|---|---|---|---|---|---|
| ***T1*** | | | | | | | | | | | | |
| ***Individual factors*** | | | | | | | | | | | | |
| Age | 0.07 | 1.07 (0.86,1.33) | 0.08 | 1.09 (0.86,1.37) | 0.11 | 1.12 (0.88,1.41) | 0.12 | 1.12 (0.88,1.43) | 0.11 | 1.12 (0.88,1.43) | 0.15 | 1.16 (0.90,1.50) |
| Gender | | | | | | | | | | | | |
| Male | | 1 | | 1 | | 1 | | 1 | | 1 | | 1 |
| Female | -0.18 | 0.84 (0.64,1.10) | -0.21 | 0.82 (0.61,1.08) | -0.21 | 0.81 (0.61,1.08) | -0.21 | 0.82 (0.60,1.10) | -0.21 | 0.81 (0.60,1.10) | -0.26 | 0.77 (0.57,1.05) |
| Siblings | | | | | | | | | | | | |
| One child | | 1 | | 1 | | 1 | | 1 | | 1 | | 1 |
| Non-one child | -0.01 | 0.99 (0.75,1.31) | 0.17 | 1.19 (0.87,1.63) | 0.23 | 1.26 (0.92,1.74) | 0.22 | 1.25 (0.90,1.74) | 0.21 | 1.24 (0.89,1.73) | 0.19 | 1.21 (0.86,1.69) |
| Migrant status | | | | | | | | | | | | |
| Migrant students | | 1 | | 1 | | 1 | | 1 | | 1 | | 1 |
| Local students | 0.37 | 1.44 (1.01,2.05)* | 0.24 | 1.27 (0.88,1.83) | 0.29 | 1.34 (0.92,1.95) | 0.30 | 1.35 (0.92,1.99) | 0.31 | 1.37 (0.93,2.02) | 0.41 | 1.51 (1.01,2.25)* |
| Positive Youth Development (PYD) | | | | | | | | | | | | |
| Cognitive Behavioral Competence | -0.29 | 0.75 (0.56,1.01) | -0.28 | 0.76 (0.56,1.03) | -0.30 | 0.74 (0.54,1.02) | -0.24 | 0.79 (0.57,1.09) | -0.23 | 0.80 (0.58,1.11) | -0.25 | 0.78 (0.55,1.10) |
| Prosocial Attributes | 0.03 | 1.03 (0.83,1.29) | 0.03 | 1.03 (0.82,1.30) | -0.05 | 0.95 (0.75,1.21) | -0.04 | 0.96 (0.76,1.23) | -0.04 | 0.96 (0.75,1.22) | -0.07 | 0.93 (0.72,1.19) |
| Positive Identity | 0.36 | 1.43 (1.13,1.83)** | 0.36 | 1.43 (1.11,1.83)** | 0.30 | 1.35 (1.05,1.75)* | 0.23 | 1.26 (0.97,1.63) | 0.23 | 1.26 (0.97,1.64) | 0.27 | 1.31 (1.00,1.72) |
| General PYD Qualities | 0.92 | 2.52 (1.72,3.67)*** | 0.61 | 1.85 (1.24,2.74)** | 0.49 | 1.63 (1.09,2.44)* | 0.40 | 1.50 (0.99,2.26) | 0.41 | 1.50 (0.99,2.28) | 0.47 | 1.61 (1.04,2.48)* |
| ***Family factors*** | | | | | | | | | | | | |
| Family intactness | | | | | | | | | | | | |
| Intactness | | | | 1 | | 1 | | 1 | | 1 | | 1 |
| Divorced parents | | | -0.04 | 0.97 (0.37,2.56) | 0.09 | 1.09 (0.41,2.90) | 0.30 | 1.35 (0.49,3.74) | 0.32 | 1.37 (0.49,3.87) | 0.41 | 1.50 (0.52,4.37) |
| Single parent family | | | -0.06 | 0.94 (0.30,2.97) | -0.11 | 0.90 (0.28,2.84) | 0.11 | 1.11 (0.32,3.82) | 0.15 | 1.17 (0.34,3.99) | 0.28 | 1.32 (0.39,4.46) |
| Others | | | -0.08 | 0.92 (0.09,9.60) | -0.61 | 0.54 (0.05,5.52) | -0.59 | 0.55 (0.05,5.90) | -0.56 | 0.57 (0.05,6.20) | -0.37 | 0.69 (0.06,8.38) |
| Per capita monthly income in family (RMB) | | | | | | | | | | | | |
| <1,000 | | | | 1 | | 1 | | 1 | | 1 | | 1 |
| 1,000–1,999 | | | 0.22 | 1.24 (0.40,3.87) | 0.28 | 1.33 (0.42,4.21) | 0.19 | 1.21 (0.36,4.03) | 0.21 | 1.24 (0.37,4.15) | 0.15 | 1.16 (0.33,4.06) |

(*Continued*)

**Table 5.** (*Continued*)

| Predictors | Model 1 B | Model 1 OR and 95% C.I. for OR | Model 2 B | Model 2 OR and 95% C.I. for OR | Model 3 B | Model 3 OR and 95% C.I. for OR | Model 4 B | Model 4 OR and 95% C.I. for OR | Model 5 B | Model 5 OR and 95% C.I. for OR | Model 6 B | Model 6 OR and 95% C.I. for OR |
|---|---|---|---|---|---|---|---|---|---|---|---|---|
| 2,000–2,999 | | | 0.23 | 1.26 (0.43,3.73) | 0.37 | 1.45 (0.48,4.37) | 0.26 | 1.30 (0.41,4.11) | 0.30 | 1.35 (0.42,4.29) | 0.18 | 1.19 (0.36,3.98) |
| 3,000–3,999 | | | 0.15 | 1.16 (0.40,3.40) | 0.27 | 1.31 (0.44,3.89) | 0.22 | 1.25 (0.40,3.90) | 0.24 | 1.27 (0.40,3.98) | 0.13 | 1.14 (0.35,3.75) |
| 4,000–4,999 | | | 0.54 | 1.71 (0.58,5.06) | 0.63 | 1.88 (0.63,5.66) | 0.52 | 1.68 (0.53,5.30) | 0.52 | 1.68 (0.53,5.34) | 0.30 | 1.35 (0.40,4.47) |
| 5,000–5,999 | | | 0.29 | 1.33 (0.45,3.97) | 0.39 | 1.48 (0.49,4.50) | 0.34 | 1.41 (0.44,4.49) | 0.37 | 1.45 (0.45,4.64) | 0.14 | 1.15 (0.34,3.88) |
| ≥6,000 | | | 0.39 | 1.48 (0.52,4.17) | 0.55 | 1.73 (0.60,4.97) | 0.44 | 1.55 (0.51,4.67) | 0.47 | 1.61 (0.53,4.88) | 0.32 | 1.38 (0.43,4.39) |
| Father's Education Level | | | | | | | | | | | | |
| Middle school or lower | | | | 1 | | 1 | | 1 | | 1 | | 1 |
| High school or college | | | 0.17 | 1.18 (0.82,1.71) | 0.23 | 1.26 (0.87,1.84) | 0.21 | 1.23 (0.83,1.81) | 0.20 | 1.22 (0.83,1.80) | 0.13 | 1.14 (0.77,1.70) |
| Graduate | | | 0.45 | 1,58 (0.92,2.70) | 0.50 | 1.65 (0.96,2.86) | 0.39 | 1.48 (0.84,2.62) | 0.39 | 1.48 (0.83,2.62) | 0.42 | 1.52 (0.85,2.75) |
| Above graduate | | | 0.27 | 1.31 (0.65,2.62) | 0.33 | 1.39 (0.68,2.83) | 0.18 | 1.20 (0.60,2.51) | 0.17 | 1.18 (0.56,2.49) | 0.17 | 1.19 (0.55,2.57) |
| Mother's Education Level | | | | | | | | | | | | |
| Middle school or lower | | | | 1 | | 1 | | 1 | | 1 | | 1 |
| High school or college | | | 0.07 | 1.07 (0.74,1.55) | 0.04 | 1.04 (0.71,1.51) | 0.04 | 1.04 (0.71,1.54) | 0.03 | 1.03 (0.70,1.51) | 0.07 | 1.07 (0.72,1.59) |
| Graduate | | | -0.01 | 0.99 (0.58,1.71) | -0.01 | 0.99 (0.57,1.74) | -0.08 | 0.93 (0.52,1.64) | -0.08 | 0.92 (0.52,1.64) | -0.15 | 0.86 (0.48,1.55) |
| Above graduate | | | -0.23 | 0.80 (0.36,1.76) | -0.30 | 0.74 (0.33,1.67) | -0.32 | 0.73 (0.31,1.70) | -0.35 | 0.71 (0.30,1.66) | -0.36 | 0.70 (0.29,1.70) |
| Family Function | | | | | | | | | | | | |
| Family mutuality | | | -0.10 | 0.90 (0.71,1.16) | -0.09 | 0.91 (0.71,1.17) | -0.09 | 0.92 (0.71,1.19) | -0.08 | 0.92 (0.71,1.19) | -0.08 | 0.93 (0.71,1.21) |
| Family harmony | | | 0.25 | 1.29 (1.04,1.59) * | 0.20 | 1.22 (0.98,1.51) | 0.18 | 1.20 (0.96,1.49) | 0.16 | 1.17 (0.94,1.46) | 0.15 | 1.16 (0.93,1.45) |
| Family communication | | | 0.36 | 1.43 (1.18,1.75) *** | 0.36 | 1.43 (1.17,1.75) ** | 0.38 | 1.47 (1.19,1.80) *** | 0.37 | 1.44 (1.17,1.78) ** | 0.37 | 1.45 (1.17,1.80) ** |
| ***School factors*** | | | | | | | | | | | | |
| School Maladjustment | | | | | | | | | | | | |
| Classroom problem behaviour | | | | | 0.00 | 1.00 (0.89,1.12) | 0.03 | 1.03 (0.91,1.16) | 0.02 | 1.02 (0.91,1.16) | 0.04 | 1.04 (0.92,1.18) |
| Academic fraud | | | | | -0.02 | 0.98 (0.90,1.07) | -0.01 | 0.99 (0.90,1.08) | -0.01 | 0.99 (0.90,1.08) | -0.01 | 0.99 (0.91,1.09) |
| Awareness of no expectation from teacher | | | | | -0.15 | 0.86 (0.79,0.94) ** | -0.14 | 0.87 (0.79,0.95) ** | -0.14 | 0.87 (0.79,0.95) ** | -0.13 | 0.88 (0.80,0.97) ** |

(*Continued*)

**Table 5.** (Continued)

| Predictors | Model 1 | | Model 2 | | Model 3 | | Model 4 | | Model 5 | | Model 6 | |
|---|---|---|---|---|---|---|---|---|---|---|---|---|
| | B | OR and 95% C.I. for OR | B | OR and 95% C.I. for OR | B | OR and 95% C.I. for OR | B | OR and 95% C.I. for OR | B | OR and 95% C.I. for OR | B | OR and 95% C.I. for OR |
| Poor interpersonal relationship | | | | | -0.14 | 0.87 (0.78,0.97) * | -0.17 | 0.84 (0.75,0.95) ** | -0.17 | 0.85 (0.75,0.95) ** | -0.17 | 0.84 (0.75,0.95) ** |
| **T2** | | | | | | | | | | | | |
| **Individual factors** | | | | | | | | | | | | |
| Positive Youth Development (PYD) | | | | | | | | | | | | |
| Cognitive Behavioral Competence | | | | | | | -0.30 | 0.74 (0.54,1.02) | -0.32 | 0.73 (0.53,1.01) | -0.35 | 0.71 (0.50,0.99) * |
| Prosocial Attributes | | | | | | | -0.14 | 0.87 (0.66,1.14) | -0.15 | 0.86 (0.66,1.13) | -0.25 | 0.78 (0.59,1.03) |
| Positive Identity | | | | | | | 0.37 | 1.45 (1.12,1.89) ** | 0.38 | 1.46 (1.12,1.91) ** | 0.37 | 1.45 (1.10,1.92) ** |
| General PYD Qualities | | | | | | | 0.75 | 2.11 (1.45,3.06) *** | 0.66 | 1.94 (1.31,2.87) ** | 0.52 | 1.68 (1.12,2.52) * |
| **Family factors** | | | | | | | | | | | | |
| Family Function | | | | | | | | | | | | |
| Family mutuality | | | | | | | | | -0.08 | 0.92 (0.72,1.18) | -0.10 | 0.90 (0.70,1.16) |
| Family harmony | | | | | | | | | 0.17 | 1.18 (0.97,1.45) | 0.18 | 1.19 (0.97,1.47) |
| Family communication | | | | | | | | | 0.12 | 1.13 (0.92,1.39) | 0.15 | 1.16 (0.93,1.43) |
| **School factors** | | | | | | | | | | | | |
| School **Maladjustment** | | | | | | | | | | | | |
| Classroom problematic behaviour | | | | | | | | | | | 0.02 | 1.02 (0.90,1.16) |
| Academic fraud | | | | | | | | | | | -0.07 | 0.93 (0.86,1.01) |
| Awareness of no expectation from teacher | | | | | | | | | | | -0.11 | 0.90 (0.82,0.98) * |
| Poor interpersonal relationship | | | | | | | | | | | -0.20 | 0.82 (0.73,0.92) ** |
| χ2 | 108.65*** | | 146.45*** | | 179.56*** | | 234.67*** | | 239.57*** | | 282.52*** | |
| -2 × log likelihood | 1238.74 | | 1200.94 | | 1167.83 | | 1112.71 | | 1107.81 | | 1064.86 | |
| Cox & Snell R2 | 0.10 | | 0.14 | | 0.16 | | 0.21 | | 0.21 | | 0.24 | |
| Nagelkerke R2 | 0.14 | | 0.18 | | 0.22 | | 0.28 | | 0.29 | | 0.33 | |

*p <0.05

**p <0.01

***p <0.001

to depression in adolescents, which help to draw the outline of developmental trend of non-depression in adolescents in junior middle school.

Based on the ecological system theory, risk and protective factors of persistent non-depressive symptoms were examined from the individual, familial and school levels in our study. At the individual level, local student was expected to reduce persistent depression and increase persistent non-depression, since those local students may be more familiar and adaptive with local life and local people which might be a capital when they are challenged with family or school life. However, this was only confirmed for persistent non-depression at 1st-year and 2nd-year follow-up. The results indicate that been local student is not a protective factor for persistent depression of adolescents, but it promotes adolescents non-depression longitudinally. In addition, we found that higher scores of personal positive identity and general positive youth development qualities significantly increased the persistence of non-depression in the second and third year, indicating that the effect of positive youth development (PYD) on the persistent non-depression would be lasting. In addition, personal positive identity reduced persistent depression only at 2nd-year follow-up. Interestingly, prosocial attributes increased persistent-depression and personal cognitive-behavioral competence decreased non-depression of adolescents, which suggested that some prosocial attribute may be against adolescents' id which may be more destroyable, and good cognitive-behavioral competence may lead to higher goals which triggers more depression. In total, PYD, defined as the growth, cultivation, and nurturance of developmental assets, abilities, and potentials in adolescents [45], mainly have a positive effect on academic achievement and psychological adjustment in youth adolescence [46], which could partially explain the role it played in the persistent depression/non-depression in our study.

At the family level, it was very interesting to find that psychosocial factors like family intactness and family income were not associated with persistent depression or persistent non-depression, and father's education level protected adolescents from persistent depression only at 1st-year follow up, which was inconsistent with the previous studies on psychosocial correlates of adolescent depression. According to a national household survey in China, adolescents were more likely to report depressive symptoms when family income decreased [7]. An European study investigated the independent effect of parental education levels in a sample of children (between 8 and 11 years of age) and adolescents (between 12 and 18 years of age), and results showed that low parental education predicted lower scores on psychological wellbeing, moods and emotions in children, but not in adolescents [47]. Therefore, disparities exist in different studies and it is possible that the percentages of adolescents in different family structure and family income in our study were very close between depression and non-depression groups at baseline, and persistent non-depression group reported higher education level of father, leading to the above different results. Furthermore, other more influential factors of persistent depression/non-depression may exist beyond our study. As one study revealed that an incomplete family structure would do harm to the growth of adolescents but a good parent-child relationship could compensate for the harm that non-intact family generated [24]. From this perspective, we can infer that the mechanism and psychosocial factors behind the persistent depression/non-depression are more complex than expected.

We also found that having a higher level of family functioning at baseline decreased the persistent depression and increased the persistent non-depression in the second and third year. Previous studies on the association of family functioning with adolescent depression showed that good family functioning was associated with less depression in Chinese adolescents [48, 49]. Family functioning was assessed by three domains in our study, which were parent-child mutuality, family harmony and conflict, and communication. Among them, family harmony and good communication were significantly regarded as protective factors of persistent

depression in the second and third year. Family conflict had a negative effect on adolescents physically and mentally, especially when the conflict was associated with them, then adolescents might have a feeling of helplessness and guilty towards the family conflict they could not avoid being involved in, even lead to cognitive dissonance and lack of adequate skills to deal with interpersonal communication, and develop into depression finally [24]. Harmonious family does not mean one family without conflicts but conflicts can be solved well in it, so adolescents from harmonious families are less likely to suffer from depressive symptoms than those from families with severe conflicts. As to family communication, our results indicated that good communication in family consistently promoted non-depression at Wave 2 and Wave 3. Most domestic previous studies showed that depressive adolescents had poor family communication and lack of emotion exchange with their parents, so good family communication between adolescents and parents would promote emotion exchange with each other, thus reducing the likelihood of adolescent depression. To summarize, family functioning is a protective psychosocial factor of persistent depression and a promoter of persistent non-depression. Measures should be taken to improve parent-child relationship, family communication as well as family harmony to protect adolescents from developing depressive symptoms.

At the school level, adolescents who had a range of maladaptive behaviors such as problematic behaviors in class, academic fraud, negative perception of teacher's expectation and poor interpersonal relationship with peers or teachers, were considered as school maladjustment [37]. We found that negative perception of teacher's expectation and poor interpersonal relationship with peers or teachers at baseline were significant risk factors for the persistence of non-depression in the second and third year. Holding higher expectation (e.g., "expect me to work hard in school") from teachers was regarded as a form of teacher support at school, relating to a higher hope and lower psychosocial distress for adolescents, and less stress and more support were associated with fewer mental health problems [50]. Therefore, having a negative perception of teacher's expectation was a risk factor of persistent non-depression among Chinese adolescents. Furthermore, adolescents having poor interpersonal relationships at school were more likely to suffer from depressive symptoms but less likely to have non-depressive symptoms in our study, which was supported by one survey that poor interpersonal relationships increased the susceptibility to depression among both urban and rural adolescents [51]. In all, school maladjustment is a significant risk factor of persistent non-depressive symptoms and school-based depression prevention programs should be cultivated in future. Together, individual, family and school protective and risk factors were indicated for persistent depression symptoms (Hypothesis 4) and non-depression symptoms (Hypothesis 5). The results enrich the ecological system theory in adolescent depression and non-depression, i.e., individual, family and school protective and risk predictors contribute to adolescent depression and non-depression together. This knowledge helps to prevent depression and promote non-depression in youth.

The present study had several limitations: First, all data were only self-reported by adolescents themselves, we need to be cautious in explanation of these results, considering their cognitive immaturity and inaccuracy report of information. In addition, the adolescents were selected only from Shenzhen, southern China. It should be cautious to generalize the results from our study to other regions of China. However, strengths also existed in our study. First, the longitudinal design was applied in our study, and we traced the trend of depression in Chinese early adolescents over three-years. Second, unlike the existing literature mostly targeted at adolescents with depressive symptoms, we paid extra attention to those adolescents without depressive symptoms and our study provided a complete data on the adolescent depression. Lastly, we explored psychosocial factors related to the adolescent persistent non-depression based on the ecological systems theory [9], which provided a scientific theoretical framework

to analyze psychosocial factors in our study. Future studies are encouraged to conduct a larger sample longitudinal survey combined with interview as well as parents and teacher investigation to further enhance the power of the findings about adolescent depression and non-depression.

## Conclusions

In summary, this study reveals that the incidence of depressive symptoms is not high in the sample of adolescents aged 11–15 years-old in junior middle school. Secondly, the rate of persistent non-depression among Chinese adolescents is approximately 30% higher than that of persistent depression longitudinally, and both have a downward trend over years, suggesting the unnecessary to be panic and much attention should be paid to persistent non-depression in adolescents who may possess higher level of assets against depression. Furthermore, positive youth developmental assets at the individual level and family functioning at the familial level are protective factors of persistent non-depression, while school maladjustment is a risk factor, indicating that future adolescent depression prevention would be beneficial through targeting at promoting adolescents' positive youth development, family functioning and school adjustment.

## Supporting information

**S1 Table. Comparison between dropouts and formal sample.**
(DOCX)

**S1 Data. Raw data of this work.**
(ZIP)

## Acknowledgments

The authors would like to thank all participating junior secondary school teachers in Shenzhen for their help with data collection and all students for their voluntarily and positively participating in the study.

## Author Contributions

**Data curation:** Xin-li Chi.

**Formal analysis:** Ming Liu, Fei Xie, Zongpei Dai.

**Project administration:** Qin Dai, Xin-li Chi.

**Supervision:** Qin Dai, Xin-li Chi.

**Writing – original draft:** Fei Xie.

**Writing – review & editing:** Ming Liu, Qin Dai.

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
