## [Decision Letter · Decision Letter 0]

3 Jul 2024

PONE-D-24-21487Trend and Predictive Psychosocial Factors of Persistent Depression/Non-depression in Chinese Adolescents: a Three-year Longitudinal StudyPLOS ONE

Dear Dr. Dai,

Thank you for submitting your manuscript to PLOS ONE. After careful consideration, we feel that it has merit but does not fully meet PLOS ONE’s publication criteria as it currently stands. Therefore, we invite you to submit a revised version of the manuscript that addresses the points raised during the review process.

We look forward to receiving your revised manuscript.

Kind regards,

Muddsar Hameed

Academic Editor

PLOS ONE

Journal Requirements:

2. Thank you for stating the following financial disclosure: "This research was funded by the National Social Science Foundation of China (16CSH049) and key project of natural science foundation of Chongqing (cstc2020jcyj-zdxmX0009)."

3. In the online submission form, you indicated that all data generated or analyzed during this study are available from corresponding author (daiqin101@hotmail.com) through adequate request.

Reviewers' comments:

Reviewer's Responses to Questions

**Comments to the Author**

1. Is the manuscript technically sound, and do the data support the conclusions?

Reviewer #1: Yes

Reviewer #2: Yes

2. Has the statistical analysis been performed appropriately and rigorously? 

Reviewer #1: Yes

Reviewer #2: Yes

3. Have the authors made all data underlying the findings in their manuscript fully available?

Reviewer #1: Yes

Reviewer #2: No

4. Is the manuscript presented in an intelligible fashion and written in standard English?

Reviewer #1: Yes

Reviewer #2: Yes

5. Review Comments to the Author

Reviewer #1: 1. The introduction emphasizes the significance of researching non-depressive symptoms; however, it might provide additional details regarding the special nature of this topic and its potential benefits for the field of adolescent mental health.

2. Rationale of the study should be included

3. The ecological system theory is mentioned as a framework, but there is little discussion on how this theory has been applied in previous research on adolescent depression.

4. A theoretical framework could have been drawn as a support/guidance

5. While the attrition rate of 15.7% is mentioned, there is no detailed analysis of the characteristics of participants who dropped out compared to those who remained. This could help understand if attrition biased the results.

6. Include a comparison of demographic and baseline characteristics between completers and dropouts to assess potential attrition bias.

7. The sample could have been drawn from schools of various socio-economical status to increase the generalizability of the results

8. Seasonal effect on the depression (winter depression) could have been mentioned as it can act as a confounding factor.

9. Elaborate on how ecological system theory informed the study design, variable selection, and the interpretation of findings.

10. Address how potential confounding variables were accounted for in the analysis, ensuring that the observed effects are not due to unmeasured factors

11. Clearly define each group with specific criteria and thresholds used for classification, ensuring that the reader can easily understand the distinctions such as mixed depression

12. Elaborate on the role of migrant status in the context of depressive symptoms, providing insights into why this might be a significant factor.

Reviewer #2: sample size is justified for a study measuring prevalence and attrition is well-managed.

the stats from UNICEF in the first line need to be given with complete reference with year and date so that rationale can be stronger.

rationale for a longitudinal study is made logically however they need to keep in mind large sample size of previous studies.

how is gender correlated with depression? Please rephrase in line 90

most factors that are protective of depression have been explained in lines 90-117 and seem without any confusion thus rationale for finding protective factors gets weak, try to mention what was missing or what are the methodological gaps in the literature that prompted you to study this

how were you able to manage the demand characteristics due to self-report of data?

procedure, and results are well written along with the statistical analysis.

in discussion the hypothesis 1 results are not completely explained. reasoning needs to be added, are lower depression rates related to the presence of protective factors, what is common among the sample that's causing them to have lower depression rates?

rephrase 321-322

rest of discussion is well written and reasoning is appropriate,

limitation of self-report is acknowledged, due to which caution should be added in interpreting results.

future directions are not mentioned which need to be added.

6. PLOS authors have the option to publish the peer review history of their article (what does this mean?). If published, this will include your full peer review and any attached files.

Reviewer #1: **Yes: **Khazeneen Atiq

Reviewer #2: No

---

## [Author Response · Author response to Decision Letter 0]

18 Jul 2024

Dear Dr. Muddsar Hameed,

Thank you for giving us an opportunity to revise the manuscript. Please find enclosed revised version of our manuscript entitled “Trend and Predictive Psychosocial Factors of Persistent Depression/Non-depression in Chinese Adolescents: a Three-year Longitudinal Study” for your consideration.

Many thanks for your constructive and detailed comments on the manuscript, which substantially improves the paper. In this revision version, we have addressed all of comments made by the reviewers (see below). With these changes, we hope that the manuscript will be of more relevant to the readership of your journal.

This manuscript is not in consideration elsewhere and the author reports no conflicts of interest. Moreover, the co-author has approved the revised version making significant contributions to the writing and conceptualization of the manuscript.

If I can be of any further assistance, please do not hesitate to contact me at my email address (daiqin101@hotmail.com). Thank you for your consideration.

Yours sincerely,

Corresponding author

Reviewer's comments to the Author:

Journal Requirements:

Author’s responding: Thank you for your detailed suggestion and revised accordingly. This version meets PLOS ONE's style requirements. See manuscript for details.

2. Thank you for stating the following financial disclosure: "This research was funded by the National Social Science Foundation of China (16CSH049) and key project of natural science foundation of Chongqing (cstc2020jcyj-zdxmX0009)."

Author’s responding: Thank you for your detailed suggestion and revised accordingly. See Declarations Funding for details: “This research was funded by the key project of natural science foundation of Chongqing (cstc2020jcyj-zdxmX0009), Humanities and Social Sciences of Ministry of Education Planning Foundation (Grant 23YJA190002), and Guangdong Philosophy and Social Science Planning 2023 Co-construction Project (Grant GD23XXL17). The funders had a role in study design, data collection and decision to publish, while they had no role in data analysis and preparation of the manuscript.”

3. In the online submission form, you indicated that all data generated or analyzed during this study are available from corresponding author (daiqin101@hotmail.com) through adequate request.

Author’s responding: Thank you for your detailed suggestion and revised accordingly. We uploaded data as supplementary information.

Author’s responding: Thank you for your detailed suggestion and revised accordingly. We deleted ethics statement in Declarations and described it at Methods section. See 2.3 Procedure for details: “The study and data collection received approval from the surveyed schools and the Human Research Ethics Committee of Shenzhen University. All methods were performed in accordance with the relevant guidelines and regulations. All students were informed of the study purpose and their data would be analyzed and published with personal information kept in strict confidentiality. Before the data collection, the written informed consent of participants was obtained from both students and their parents……”

Author’s responding: Thank you for your detailed suggestion and revised accordingly. Reference 36 was deleted, references 2, 17, 27, 35 were replaced, references 8, 11, 12, 30, 44 were added, and Chinese references were indicated. Moreover, the first six authors were listed (followed by et al.) as well as all available doi were added. See References for details.

Reviewers' comments:

Reviewer #1: 

Author’s responding: Thank you for your valuable suggestions, which substantially improves the paper. 

1. The introduction emphasizes the significance of researching non-depressive symptoms; however, it might provide additional details regarding the special nature of this topic and its potential benefits for the field of adolescent mental health.

Author’s responding: Thank you for your valuable suggestion and revised accordingly. Details regarding the special nature of this topic and its potential benefits for adolescent mental health were added in Discussion. See the second paragraph of Introduction for details: “……Although experiencing same adversity, same people suffer from depression, while most do not suffer from depression but manifest certain resilience, which could be used to prevent and cure depression. Adolescents without depressive symptoms may possess different personal diathesis (such as resilience) [8] and demographic-social characteristics (e.g., higher socio-economical status) which protect them from depressed mood……”

2. Rationale of the study should be included

The ecological system theory is mentioned as a framework, but there is little discussion on how this theory has been applied in previous research on adolescent depression.

A theoretical framework could have been drawn as a support/guidance

Elaborate on how ecological system theory informed the study design, variable selection, and the interpretation of findings.

Author’s responding: Thank you for your inspiring suggestion and revised accordingly. The ecological system theory in adolescent depression and a theoretical framework were added in Introduction. And how ecological system theory informed the study design, variable selection, and the interpretation of findings, were further added in Introduction and Discussions. See the third and seventh paragraph of Introduction “Apart from the prevalence of adolescent depression, researchers also have been exploring risk and protective factors of adolescent depressive symptoms. Based on the ecological system theory [10], personal and environmental factors (family and school, et al.) contributed to mental health status together. Based on recent reviews, researchers have highlighted the integral role of ecological factors to the American National Institute of Mental Health Research Domain Criteria framework in predicting onset and maintenance of internalizing problems in youth [11]. Furthermore, the risk factors and protective factors of young people relating to the mental health outcomes of both direct and indirect exposure to climate change were reviewed through the lens of ecological system theory [12]. Thus, for adolescents, specifically, it could be hypothesized that individual, family, and school factors may predict the onset of their depression together (Fig 1). Considering the studies on adolescents without depressive symptoms are sparse, information about the factors contributing to non-depression in adolescents are even less, so we intended to explore psychosocial predictors (individual, family, and school factors) of persistent depression/non-depression in adolescents based on the theoretical framework of the ecological system theory [13], which is potentially important to figure out a way to better prevent adolescents from depression……Building upon the existing literatures about adolescent depression, although factors from individual, family, and school levels were indicated separately, however, most studies only focused on solely one domain of the ecological system, and most of them were cross-sectional design. Specifically, some variables under Chinese culture context (one-child policy, migrant population between provinces) [30] did not include in previous survey. Thus, their combined effect on longitudinal prediction of depression in Chinese adolescent remains unclear. Notably, predictors of adolescent non-depression from ecological system are unexplored.” and third and sixth paragraph of Discussions “Based on the ecological system theory, risk and protective factors of persistent non-depressive symptoms were examined from the individual, familial and school levels in our study. At the individual level……The results enrich the ecological system theory in adolescent depression and non-depression, i.e., individual, family and school protective and risk predictors contribute to adolescent depression and non-depression together. This knowledge helps to prevent depression and promote non-depression in youth” for details.

3. While the attrition rate of 15.7% is mentioned, there is no detailed analysis of the characteristics of participants who dropped out compared to those who remained. This could help understand if attrition biased the results.

Include a comparison of demographic and baseline characteristics between completers and dropouts to assess potential attrition bias.

Author’s responding: Thank you for your valuable suggestion and revised accordingly. A comparison of demographic and baseline characteristics between completer and dropouts were applied. See 2.1. Participants and Supplementary tables for details: “……Comparison between 243 dropouts and 1301 formal sample found no significant differences on demographic and baseline characteristics between them, which indicated that attrition did not bias the results ( S1 Table)……” 

 S1 Table Comparison between dropouts and formal sample 

 Drop-out (n = 243) Final sample (n = 1301) F/x2

Gender 2.438

Male 139 666 

Female 104 621 

Missing data 14 

Siblings 3.074

One Child 79 499 

Non-one child 164 799 

Missing data 3 

Migrant status 2.692

Migrant students 55 235 

Local students 188 1060 

Missing data 6 

Positive Youth Development 0.509

Family intactness 1.444

Intactness 232 1222 

Divorced parents 7 36 

Non-Intactness 2 24 

Others 2 8 

Missing data 11 

Per capita monthly income in family (RMB) 4.201

＜1,000 3 27 

1,000-1,999 21 81 

2,000-2,999 25 135 

3,000-3,999 41 169 

4,000-4,999 24 121 

5,000-5,999 33 127 

≥6,000 88 439 

Missing data 8 202 

Father’s Education Level 3.814

Middle school or lower 84 404 

High school or college 101 451 

Graduate 41 226 

Above graduate 16 120 

Minssing data 1 100 

Mother’s Education Level 1.437

Middle school or lower 99 492 

High school or college 90 427 

Graduate 36 217 

Above graduate 13 74 

Missing data 5 91 

4. The sample could have been drawn from schools of various socio-economical status to increase the generalizability of the results

Author’s responding: Thank you for your valuable suggestion and revised accordingly. The socio-economical status of school were added in Methods and further discussed in Discussions. See 2.1. Participants: “……all of which belonged to medium socio-economical status……” and the third paragraph of Discussion for details: “……Moreover, all six schools belonged to medium socio-economical status. However, Table 1 suggested that students came from families with different socio-economical statuses, with approximate 40% per capita monthly income in family (RMB) ≥6,000 and approximate 10% < 2000, which could represent students from low, medium, and high socio-economical statuses of families. Future studies are encouraged to include school of various socio-economical statuses to guarantee broader generalization of the results……”

5. Seasonal effect on the depression (winter depression) could have been mentioned as it can act as a confounding factor.

Author’s responding: Thank you for your important suggestion and revised accordingly. Seasonal effect on the depression was mentioned in the discussion. See the first paragraph of Discussion for details: “……Notably, seasonal affective disorder is a mood disorder that is characterized by depressive symptoms that occur at a specific time of year (typically fall or winter) with full remission at other times of year (typically spring or summer). The investigation time-points of this study were October to November during three successive years, which may be potentially influenced by the season change. Future study may choose other survey season to further verify the results.”

6. Address how potential confounding variables were accounted for in the analysis, ensuring that the observed effects are not due to unmeasured factors

Author’s responding: Thank you for your important suggestion and revised accordingly. See the second paragraph of 2.4. Statistical analyses for details: “……In which, potential confounding variable were controlled at level 1 (baseline age, gender, whether have siblings, migrant status) and level 2 (baseline father’s and mother’s educational level, family intactness or not, monthly household income)……”

7. Clearly define each group with specific criteria and thresholds used for classification, ensuring that the reader can easily understand the distinctions such as mixed depression

Author’s responding: Thank you for your important suggestion and revised accordingly. See 2.4. Statistical analyses for details: “According to the three-wave CES-D scores in our study, three groups were identified[30]: 1) Persistent depression symptom, in which adolescents were constantly classified as reporting symptoms (CES-D ≥16) from Wave 1 to 3; 2) Persistent non-depression, in which adolescents report no depression symptoms (CES-D <16) continuously from T1 to T3; 3) Mixed depression symptom, in which adolescents were classified as reporting symptoms (CES-D ≥16) once (i.e., mixed depression across 12 months) or no more than twice (i.e., mixed depression across 24 months) from Wave 1 to 3”.

8. Elaborate on the role of migrant status in the context of depressive symptoms, providing insights into why this might be a significant factor.

Author’s responding: Thank you for your detailed suggestion and revised accordingly. See the third paragraph of Discussion for details: “……At the individual level, local student was expected to reduce persistent depression and increase persistent non-depression, since those local students may be more familiar and adaptive with local life and local people which might be a capital whe

---

## [Editor Report · Decision Letter 1]

22 Jul 2024

Trend and Predictive Psychosocial Factors of Persistent Depression/Non-depression in Chinese Adolescents: a Three-year Longitudinal Study

PONE-D-24-21487R1

Dear Dr. Qin Dai ,

We’re pleased to inform you that your manuscript has been judged scientifically suitable for publication and will be formally accepted for publication once it meets all outstanding technical requirements.

Kind regards,

Muddsar Hameed

Academic Editor

PLOS ONE

---

## [Editor Report · Acceptance letter]

21 Aug 2024

PONE-D-24-21487R1 

PLOS ONE

Dear Dr. Dai, 

I'm pleased to inform you that your manuscript has been deemed suitable for publication in PLOS ONE. Congratulations! Your manuscript is now being handed over to our production team.

Kind regards, 

on behalf of

Dr. Muddsar Hameed 

Academic Editor

PLOS ONE